# Harnessing the Antibacterial Properties of Fluoridated Chitosan Polymers against Oral Biofilms

**DOI:** 10.3390/pharmaceutics14030488

**Published:** 2022-02-23

**Authors:** Dien Puji Rahayu, Roger Draheim, Aikaterini Lalatsa, Marta Roldo

**Affiliations:** 1School of Pharmacy and Biomedical Sciences, University of Portsmouth, St Michael’s Building, White Swan Road, Portsmouth PO1 2DT, UK; dien.rahayu@port.ac.uk (D.P.R.); roger.draheim@port.ac.uk (R.D.); 2National Research and Innovation Agency of Indonesia (BRIN), Lebak Bulus Raya No. 49, Jakarta 12440, Indonesia

**Keywords:** chitosan, fluoride, antimicrobial properties, demineralization protection, dental caries

## Abstract

Dental caries are a worldwide endemic chronic disease affecting people of all ages. Due to the limitations of daily used oral hygiene products, there is an unmet need for new, effective, safe, and economic oral products. We have recently demonstrated that *N*-(2(2,6-diaminohexanamide)-chitosan (CS3H Lys) has enhanced antibacterial properties against *Streptococcus mutans,* the main cariogenic bacterium, and here we investigated the effect of fluoridation of this polymer (CS3H Lys F) on its antibacterial properties and the ability to protect teeth from acid demineralization. We further formulated this polymer into mouthwash preparations and studied their cytocompatibility and physicochemical stability over 6 months. CS3H Lys F was 1.6-fold more effective than the highest tested oral NaF dose in preventing acid demineralization. CS3H Lys F has a 3- to 5-fold lower minimum inhibitory concentration value against *S. mutants* than the values reported for chitosan polymers and showed negligible cell toxicity. The mouthwashes were stable at both 25 and 40 °C. Further work is under way towards other CS3H Lys F oral hygiene products such as a toothpaste.

## 1. Introduction

Dental caries are a worldwide endemic chronic infection affecting people of all ages [1,2]. Almost half of the world’s population is affected by dental caries in permanent teeth [1], with particular high prevalence in young people in developing countries (79% in Thailand and 75% in Malaysia) [3,4]. Dental caries are caused by communities of microorganisms that present on the tooth surface entrapped within the extracellular polymeric substance (EPS) forming the dental biofilm or dental plaque [5,6]. The most prevalent cariogenic bacteria are *Streptococcus mutans* and *Streptococcus sobrinus* [7,8]. Enamel, the protective external layer of teeth composed of hydroxyapatite, is destroyed when the pH drops below a critical level (pH < 5.5) due to the ingestion of acidic food and drinks or the production of acid by bacteria [9]. Dental caries are preventable by (a) reducing sugar in the diet, as when metabolized by bacteria, sugar generates acid, and (b) good oral hygiene habits such as brushing teeth twice daily, combined with flossing and rinsing with a mouthwash [10]. There are several antibacterial agents such as triclosan [11,12], essential oils [13,14], cetyl pyridinium chloride [15], zinc sulphate [16], chlorhexidine (CHX) [15,16,17], or a combination of them that have been shown to control dental biofilm formation and are formulated into toothpastes and mouthwashes for ease of use and topical application [11,12,13,14,15,16,18,19,20,21]. Among these, CHX remains the gold standard for reducing plaque and gingival inflammation [22,23]. However, CHX causes tooth staining, alters taste perception and promotes calculus or tartar [17,24,25,26,27]. Thus, there is need for new, sustainable, effective, safe, and economic substances that can be formulated into daily treatment products and particularly mouthwashes that are easier to use and able to reach the narrow and small spaces in the mouth that brushing cannot.

Fluoride containing products and oral solutions remain a widely used cost-effective strategy to prevent dental caries. Fluoride ions are commonly added to oral hygiene products in the form of sodium fluoride, as they are known to disturb the growth and metabolism of cariogenic bacteria by inhibiting enolase, an enzyme involved in glycolysis, thus decreasing acid production and reducing the EPS formation in bacterial biofilms [28,29,30,31]. Additionally, fluoride ions promote remineralization of weakened tooth enamel [32,33].

Chitosan, a polysaccharide of a natural origin and a waste product of the fish industry, has been shown to possess an antibacterial activity [34,35,36], a property that justifies its inclusion in oral hygiene products. Commercially available formulations that contain chitosan as an active ingredient are available and include a chitosan-based, non-fluoride toothpaste (Chitodent^®^, B&F Elektro GmbH, Filsum, Germany), a chitosan (0.5%) toothpaste with 1400 ppm fluoride ions (F^−^) and 3500 ppm tin ions (Sn^+2^) (Elmex^®^, GABA International AG, Münchenstein, Switzerland), a chitosan argininamide mouthwash (Synedent^®^, Prisyna, Claremont, CA, USA), and a chitosan argininamide and sodium fluoride (0.05% *w*/*v* equivalent to 0.02% *w*/*v* F^−^) (Synedent FLX, Prisyna, Claremont, CA, USA). All products highlight the natural origin and low environmental impact of chitosan, supporting the wider move of many consumer health product multinational companies towards the use of naturally derived excipients and actives.

We have recently shown that *N*-(2(2,6-diaminohexanamide)-chitosan (CS3H Lys) has enhanced antibacterial properties against *S. aureus* and is able to completely inhibit its growth at concentrations as low as 200 µg mL^−1^ [37]. In this study, we demonstrate, for the first time, the effect of fluoridation of this polymer on its antibacterial properties and its ability to protect teeth from acid demineralization. We have also formulated a cytocompatible and stable (for 6 months) mouthwash readily commercialisable as an oral product.

## 2. Materials and Methods

### 2.1. Materials

Chlorhexidine gluconate (0.2%) Minosep^®^ mouthwash (Minorock Mandiri Ltd., Depok, Indonesia) and Listerine^®^ total care (Johnson and Johnson, Maidenhead, UK) were used as the control mouthwashes. 3-(4,5-dimethylthiazol-2yl)-2,5-diphenyltetrazolium bromide (MTT), trypan blue stain, penicillin/streptomycin were purchased from Fisher Scientific (Longborough, UK). All other chemicals used in this study were of analytical grade and were purchased from Sigma Aldrich Inc. (Gillingham, Dorset, UK), unless otherwise stated. CS3H was synthesized by acid degradation [38] of commercially available low-viscosity chitosan from shrimp shell (CAS 9012-96-4, Lot #BCBQ 3414V, MW: 165.3 KDa, acetylation 15.37 ± 0.47% calculated by NMR [37], Sigma-Aldrich Inc. Gillingham, Dorset, UK) and had the following properties: molecular weight 4.709 × 10^4^ g/mL, Mn 4.156 × 10^4^, Mw/Mn 1.133, acetylation 14.20 ± 0.17%, and pKa 6.68 ± 0.06. *N*-(2(2,6-diaminohexanamide)-chitosan (CS3H Lys) was synthesized from CS3H, as previously described (Appendix A) [37]. The polymer obtained had a molecular weight of 3.345 × 10^4^ g/mol (Mn 1.893 × 10^4^ and Mw/Mn 1.768), acetylation 17.56 ± 5.23%, and pKa of 6.56 ± 0.06 [37].

### 2.2. Fluoridation of Chitosan Polymers and Fluoride Quantification

Chitosan fluoride (CS3H F) was obtained by the dialysis of chitosan (1 g) against 1 L of sodium fluoride solution containing 362.5, 725, and 1450 mg NaF to obtain CS3H F_low_, CS3H F_medium_, and CS3H F_high_, respectively. Dialysis was carried out at room temperature with six changes over 24 h (MWCO: 12–14 KDa, Medicell Membranes Ltd., London, UK). Similarly, *N*-(2(2,6-diaminohexanamide)-chitosan fluoride (CS3H Lys F) was obtained by dialysis against 1 L of sodium fluoride (725 mg) solution immediately after CS3H Lys synthesis (1.54 g scale) [37] without the need for lyophilizing the product first. All dialysates were lyophilized and white polymer products were packaged in sealed polypropylene containers.

Fluoride ion loading was determined using a fluoride ion-selective electrode (ISE) (Orion Star A214, Thermo Scientific, Indonesia) fitted with a fluoride electrode (Thermo Scientific, UK). Distilled water was used for the preparation of samples and standard solutions. A total ionic strength adjustment buffer (TISAB) II with cyclohexylenedinitrilotetraacetate (CDTA) (Thermo Fisher Scientific, Warrington, UK) was used in the potentiometric measurements. Measurements were conducted as per the manufacturer’s instructions (Appendix A).

### 2.3. Fourier-Transform Infrared Spectroscopy (FTIR)

FTIR spectra of chitosan derivatives were obtained using a Varian FTIR spectrophotometer (Agilent Technologies, Stockport, UK). The samples were mounted onto the surface of the germanium (Ge) crystal in the attenuated total reflection (ATR) assembly. FTIR spectra were recorded in the middle infrared range (4000–500 cm^−1^) with a resolution of 4 cm^−1^ in the absorbance mode for 40 scans at room temperature.

### 2.4. Solubility Studies

The pH dependence of chitosan polymers aqueous solubility was evaluated at room temperature by turbidimetry [39]. Each polymer (50 mg) was dissolved in 10 mL of aqueous acetic acid (10% *v*/*v*) and stirred for 1 h; the pH level of the solutions was adjusted using NaOH solution (5 M). Measurements were repeated after each stepwise addition of NaOH until reaching pH 12. The transmittance of the solution was recorded on a Nicolet e-100 spectrophotometer (Thermo Fisher Scientific, Warrington, UK) using a quartz cell with an optical path length of 1 cm at λ 600 nm, and the pH was measured using an FE20 pH meter (Mettler Toledo, Greifensee, Switzerland).

### 2.5. In Vitro Inhibition of Acid Demineralisation

Mineralized surfaces were prepared as follows: hydroxyapatite (HA, 2 g) powder was suspended in acetone (200 mL), and 60 µL aliquots of the homogenous suspension were transferred to each well of a 96 well plate. The plate was shaken (50 rpm, microplate mixer SciQuip, Newtown, UK) at room temperature until the acetone was completely evaporated [40]. After drying, loose HA was removed by washing with deionized water (5×) and plates were allowed to dry overnight. Any plate showing poor coverage and cracking was discarded, and the coated plates were sealed until further use. Before each experiment, the plates were rehydrated with deionized water for 1 h. A phosphate standard calibration curve was prepared using KH_2_PO_4_ solutions with concentrations of phosphorus in the range 10–60 mg L^−1^ (or 0.32–1.94 mM). Deionized water and sodium fluoride were used as negative and positive controls, respectively. The positive control (sodium fluoride) solutions were prepared in deionized water in three different final concentrations: NaF_low_ (362.5 ppm), NaF_med_ (725 ppm), and NaF_high_ (1450 ppm). The polymer samples (200 µL, 1.0% *w*/*v*) were prepared in 0.2% acetic acid and the pH of all sample solutions was adjusted to 6.0 ± 0.1. Test solutions (200 µL) were added to each well and agitated (50 rpm) at room temperature for 30 min. After exposure to the test solutions, the wells were rinsed with deionized water (5×), and 200 µL of the erosive solution (0.1 M aqueous acetate buffer, pH 4.0, Alfa Aesar, Heysham, UK) was added to each well and agitated (50 rpm, 60 min). Aliquots (50 µL) were transferred into new microplates and mixed with 50 µL vanadomolybdate reagent for 5 min before reading the UV absorbance at 450 nm (SpectraMax i3x, Molecular Devices, Berkshire, UK).

### 2.6. Preparation of Bacterial Cultures

*Staphylococcus aureus* (ATCC 25923^TM^) was stored in Luria Bertani (LB) broth and sterile glycerol 30% (1:1) in a cryovial at −80 °C. The loop shalt of Culti-Loops™ *Streptococcus mutans* (ATCC 25175™) was cut from the handle using sterile scissors and dropped into warm Brain Heart Infusion (BHI, 0.5 to 1.0 mL, Oxoid, Basingstoke, UK) and incubated at 37 °C in a 5% CO_2_ atmosphere overnight. Bacteria were then streaked onto blood agar (Oxoid, Basingstoke, UK) plates using sterile cotton swabs and incubated at 37 °C in 5% CO_2_ for 24–48 h. Two to three single colonies were taken from the plate with a sterile loop and dispersed in 5 mL of fresh BHI to be incubated overnight at 37 °C in 5% CO_2_. The bacteria were stored at −80 °C by mixing the overnight culture and sterile glycerol 30% (1:1) in a cryovial. Before the experiment, the bacteria were transferred from −80 °C into 5 mL of fresh sterile BHI medium by a sterile tip and incubated at 37 °C (in 5% CO_2_ atmosphere for *S. mutans*) overnight until the medium reached an optical density at 600 nm (OD_600_) of 0.5.

### 2.7. Determination of Minimum Inhibitory Concentration (MIC) and Minimum Bactericidal Concentration (MBC)

Overnight bacterial suspensions (100 µL) were transferred into 5 mL of fresh sterile medium and incubated again until OD_600_ = 0.5. Serial dilutions of the polymer solutions from stock (5 mg mL^−1^ in 1% acetic acid) were tested at final concentrations 0.1, 0.2, 0.4, 0.8, 1.2, 1.6, 2.0, 3.2, and 4.0 mg mL^−1^ in medium. One hundred microliters of each polymer solution were placed into the well containing 5 µL of the bacterial suspensions. Wells containing only culture medium and bacteria were used as a negative control. Turbidity measurements were made for all the wells after 24 h of incubation at 37 °C for *S. aureus* and at 37 °C under anaerobic condition (5% CO_2_) for *S. mutans*. The MIC of each bacteria was recorded at 600 nm as the lowest concentration of each polymer that inhibited the bacterial growth, as detected by the absence of visual turbidity [41]. The MIC was determined as the lowest concentration of each polymer that restricted growth to a level below an OD_600_ of ≤ 0.05 [42]. The MBC was defined as the lowest concentration of test compounds that prevented any visible growth on agar plates. Samples of 20 µL were transferred from clear wells into LB agar plates, and they were incubated at 37 °C for 24 h for *S. aureus* and into blood agar plates for 48 h in 5% CO_2_ for *S. mutans*. All experiments were performed in triplicate.

### 2.8. Inhibition of S. mutans Biofilm Formation

One hundred microliters of *S. mutans* suspension were transferred to 5 mL of pre-warmed fresh BHI medium and incubated at 37 °C in a 5% CO_2_ to OD_600_ ≈ 0.5. This culture (100 µL) was then dispensed into 48 well plates, and to this biofilm medium (BM, 700 µL for negative control) or polymer solution (700 µL) in Biofilm Medium (BM) at different concentrations (0.1, 0.2, 0.4, 0.8, and 1.2 mg/mL) was added. Controls with 800 µL of BM without bacteria were also prepared. Preparation of BM was undertaken as previously described [43]. Plates were incubated for 24 h at 37 °C in a 5% CO_2_ atmosphere without agitation. After overnight incubation, the formation of biofilm was quantified by crystal violet assay [43].

### 2.9. Formulation of Chitosan-Based Mouthwash

The chitosan-based mouthwash formulas are summarized in Table 1. Lutrol (poloxamer 407) solution (4% *w*/*v*) was prepared in water, while polymers were dissolved in 0.5 M acetic acid. Three separate batches were prepared; all the ingredients were solubilized at room temperature and the pH of the mouthwash was adjusted to 5.5 ± 0.1 with NaOH (1 M).

### 2.10. Evaluation of Mouthwash Stability

Samples were stored at 25 or 40 °C for six months with eight sampling points (0, 3, 7, 14, 21, 30, 90, and 180 days) in 7 mL sample vials made of neutral glass (Type IB) and closed with polypropylene screw caps (Fisher Scientific, Loughborough, UK). Organoleptic properties such as color, odor, and appearance were monitored. Color intensity was measured by UV spectroscopy (Multiskan Go UV–VIS spectrophotometer, Thermo Fisher Scientific, Paisley, UK) at wavelengths ranging from 300 to 700 nm. Odor was subjectively assessed by the investigator. The pH was measured with a digital pH meter (Accumet AB150, Fisher Scientific, Loughborough, UK). For each sample, three independent measurements were performed, and data were reported as the mean of the replicates. pH values from the stability data were analyzed using a two-way analysis of variance (ANOVA) with Tukey’s and Sidak post-test with an a priori level significance of 0.05 to detect statistically significant differences between the time and temperature from the mouthwash formulations and control. Sedimentation was visually assessed after centrifuging 1 mL of each sample at 5000 rpm for 5 min [44] at room temperature using a Jouan B4i Centrifuge (Hemel Hampstead, UK). In vitro anti-biofilm activity was tested as described above, at all time-points of the stability study.

### 2.11. Biofilm Removal Efficacy

In vitro anti-biofilms activity was performed on overnight *S. mutans* biofilms on 48 well-plates. The *S. mutans* biofilms were prepared as described in Section 2.9. After overnight incubation at 37 °C in a 5% CO_2_ atmosphere without agitation, the plate was blotted on a paper towel to remove the culture media, and the wells were washed with distilled water to remove loosely bound cells, and the plate blotted again on a paper towel to remove all liquid. This step was repeated twice, and the plate was air-dried. Mouthwash (100 µL) was added into the well for 30 s with gentle shaking. The fluid was removed, washed with deionized water, and the well was air-dried before the addition of 50 µL of 0.1% crystal violet into each well. After 15 min, crystal violet was removed, and the wells were washed with deionized water twice. Acetic acid (33%, 200 µL) was added into each well before measuring the absorbance at 575 nm. Chlorhexidine 0.2% mouthwash and Listerine^®^ mouthwash were used as the positive controls and sterile water as a negative control.

### 2.12. Cytocompatibility Studies

Cell cytocompatibility was quantitatively measured by MTT assay using primary human gingival fibroblast (HGF-1 ATCC-CRL-2014) cells (P 7-8) derived from adult gingival tissue (ATCC, Middlesex, UK) [45,46]. Cells were cultured in DMEM (ATCC-30-2002) containing 10% FBS (FBS, ATCC-30-2025) and 1% penicillin/streptomycin (Fisher Scientific, Loughborough, UK). Mouthwash formulations were prepared as described above, but without adding the dye to avoid interference. In this experiment, artificial saliva [47] was used as a negative control; Listerine^®^ total care (a commercially available mouthwash) and 0.2% CHX Minosep^®^ mouthwash (a potent anti-plaque mouthwash) were used as the positive control. Briefly, fibroblasts were plated at a density of 5 × 10^3^ cells per well in 96-well microtiter plates, and after 24 h of incubation, the cells were exposed to 100 μL of treatment solution (mouthwash or control) for 30 or 60 s. Afterwards, MTT (5 mg mL^−1^) was dissolved in PBS, added to a final concentration of 0.5 mg mL^−1^ to each well, and incubated for further 4 h. After removing the MTT/medium, the purple formazan crystals were dissolved in 100 µL of DMSO and absorbance was measured at 570 nm and 690 nm as the background reference on a micro-plate reader SpectraMax i3x (Molecular Devices, Berkshire, UK).

### 2.13. Statistical Analysis

The results were expressed as the mean ± standard deviation (SD) for a triplicate at least, unless otherwise specified. Statistical significance was tested using GraphPad Prism version 8.2.1 for Windows (GraphPad Software, San Diego, CA USA, www.graphpad.com (accessed on 21 February 2022)).

## 3. Results

### 3.1. Preparation and Characterisation of Fluoridated Polymers

A preliminary study was conducted by loading fluoride ions onto CS3H by dialysis against sodium fluoride solutions (low against 362.5 µg mL^−1^, medium against 725 µg mL^−1^, and high against 1450 µg mL^−1^) for 24 h.

The fluoride loading capacity increased with the concentration of fluoride in the dialysis solution (Figure 1), while the fluoride entrapment efficiency (FEE) showed the opposite behavior. CS3H Lys was dialyzed against a 725 µg mL^−1^ NaF solution and its fluoride loading capacity was 104.26 ± 2.17 µg mL^−1^ with FEE equal to 29.29 ± 0.60%. The dependence of the water solubility of polymers on pH was investigated by a turbidity assay, in which the light transmittance was correlated with the water solubility. All polymers showed a decreasing solubility with increasing pH, as previously reported for chitosan polymers in the literature [48,49]. However, the formation of the fluoride ion significantly enhanced the solubility of chitosan (CS3H) at all pH values (Table 2). Fluoridation of CS3H Lys resulted in significantly higher solubility at a neutral pH. The FTIR spectra showed that no chemical modification of CS3H Lys occurs following dialysis with NaF (Figure 2), confirming loading is only by ionic interaction [50].

### 3.2. Inhibition of Acid Demineralisation In Vitro

Dental erosion of the tooth normally occurs by the action of acid consumed through food and drink, or the acid products generated by the bacteria present in the mouth; as a result, phosphate ions are released [51]. For this reason, the quantification of phosphate release from hydroxyapatite deposits after acid exposure can be used in vitro to quantify the demineralization of tooth enamel due to acid attack and evaluate the protective action of dental products. The ability of NaF solutions to prevent the acid triggered demineralization of the HA was shown to increase proportionally to the concentration of the ions for the NaF 362.5 and 725 µg mL^−1^ concentration samples, but the higher concentration of NaF (1450 µg mL^−1^) showed no further significant increase (Figure 3). All polymers showed significant protection against acid challenge of the hydroxyapatite surfaces, but only CS3H F_high_ presented a significantly higher effect compared to CS3H (Figure 3). CS3H Lys F exhibited the highest activity with 58.57% of inhibition of phosphate release.

### 3.3. Determination of MIC and MBC

The potential antibacterial activity of the modified chitosans was determined against *S. aureus* and *S. mutans*. All chitosan polymers were capable of inhibiting the growth of the microorganisms tested, with no significant difference between samples (Table 3).

### 3.4. Inhibition of Biofilms Formation

Polymers were further studied for their potential to inhibit *S. mutans* biofilm formation (Figure 4). All polymers were able to completely prevent biofilm formation at the highest concentration tested, with more than 70% efficacy showed at the lowest concentration (0.1 mg mL^−1^). At all concentrations, the modified chitosans were significantly more active than CS3H, with a higher activity demonstrated by the non-fluoridated lysine derivative. CS showed a dose dependent effect with very significant reduction in viability caused by increasing concentrations (*p* < 0.0001 for all comparisons apart from 800 vs. 1200 that the p value was *p* < 0.01). CS3H showed a dose dependent behavior (*p* < 0.001) up to 800 µg mL^−1^. CS3H Lys was highly effective with no statistical difference between concentrations above 200 µg mL^−1^ (*p* > 0.05), while no more decrease in viability was noticed with CS3H Lys F from 400 µg mL^−1^ (*p* > 0.05).

### 3.5. Cytocompatibility Study

The cytocompatibility of the mouthwash formulations was assessed against human gingival fibroblasts (HGF). Mouthwash formulations were compared against artificial saliva as the negative control and Listerine^®^ and CHX as the positive controls, and formulations were applied in their original concentration (Figure 5A) and diluted 1:1 with artificial saliva (Figure 5B) and applied for 30 and 60 s.

The result of this study was in agreement with previous studies that reported that CHX has dose-dependent cytotoxic effects on cultured gingival cells [52,53]. Listerine^®^ and 0.2% CHX showed cytotoxic effects on gingival fibroblasts with a mean viability of 51.1 ± 2.9% and 9.4 ± 0.7%, respectively, after exposure for 30 s, and 49.8 ± 2.8% and 7.8 ± 1.9%, respectively, after exposure for 60 s (Figure 5A). The percentage cell viability increased significantly (80% for Listerine^®^ and 50% for CHX) after dilution with artificial saliva (1:1). The increase in treatment time of mouthwashes did not induce a significant decrease in the percentage of HGF cell viability for both non-diluted and diluted-mouthwashes.

The MTT test revealed that all chitosan mouthwashes maintained a higher percentage of viable cells compared to the positive control solutions (*p* ≤ 0.0001) (Figure 5) for both undiluted and diluted mouthwashes.

### 3.6. Stability Studies of Mouthwash Formulations

The formulation and preparation of any new pharmaceutical or consumer care product necessitates adequate physical and chemical stability, as well as a microbiological profile unaltered over the period of time in storage under the influence of a variety of environmental factors, such as temperature, humidity, and light [54,55,56]. In this study, we evaluated the behavior of the formulations at 25 and 40 °C for 6 months.

#### 3.6.1. Organoleptic Properties

Visually, both mouthwashes presented a consistent color and clear appearance with no turbidity and a peppermint odor (subjective evaluation) that was unaltered for up to 180 days of storage both at 25 and 40 °C.

Color stability was also determined by UV measurements and changes to the absorbance values at λ_max_ 639 nm were evaluated. No statistical difference in absorbance was identified for samples stored at the two different temperatures (Appendix A and Table 4). Mouthwashes containing CS3H and CS3H Lys did not undergo any changes in absorbance for the duration of the experiment, while CS3H Lys F mouthwash showed an initial change that was reversed in time until the end of the experiment.

#### 3.6.2. Evaluation of pH Stability

As part of the stability study, the pH values of the different formulations were recorded over time (Table 5 and Appendix A), and the pH remained unaltered by time or temperature.

#### 3.6.3. Sedimentation

The appearance of each formulation was visually examined before and after sample centrifugation (5000 rpm for 5 min). No separation was observed for any of the formulations at either storage temperatures.

#### 3.6.4. Biofilm Removal Efficacy over Time

The effect of chitosan mouthwashes on *S. mutans* biofilms was observed using crystal violet staining for the duration of the stability study in order to ascertain that the mouthwash efficacy was maintained (Figure 6).

At time zero, for the samples stored at 25 °C, the control CHX solution was as effective as all our mouthwash preparations at reducing the formation of *S. mutans* biofilm (*p* > 0.05, Tukey’s multi-comparison test). A similar performance to CHX was maintained by CS3H Lys F after 6 months (*p* > 0.05). When stored at 40 °C, all formulations were statistically less effective compered to CHX (*p* > 0.05), however the CS3H Lys and CS3H Lys F formulations were more stable than the original chitosan (*p* < 0.05). The effect of temperature on biofilm reduction was further analyzed (Appendix A). Some statistical differences were observed, where the effect was higher at room temperature compared to the elevated temperature at specific time points, but no clear trend was identified, and, in all cases, efficacy was comparable at the two different temperatures after 6 months, confirming that the efficacy of the mouthwash formulations developed remained unaltered at both 25 and 40 °C for 6 months.

## 4. Discussion

Mouthwashes are designed to enhance daily oral hygiene routines by helping to minimize the formation of biofilm, and to prevent and control gingivitis, bad breath, and tooth decay. Their action is generally aided by the presence of antibacterial agents, sodium fluoride, and essential oils. Currently, chlorhexidine is the antibacterial gold standard, despite its disadvantages, such as tooth discoloration, promotion of calculus, and alteration of taste perception. Research to find new compounds with a high protective effect and anti-bacterial properties but lower toxicity than CHX is timely. It has been previously shown that positively charged chitosan chains have the ability to form protective layers on the tooth surface [57], so we combined this property with the remineralizing activity of fluoride ions, thus creating fluoridated chitosan compounds that have been prepared for the first time, to the best of our knowledge. Other studies have investigated the effect of the addition of chitosan in solution or as part of a paste formulation, and found that, while chitosan enhances the effects of fluoride compounds, it often does not have an antimicrobial effect in these formulations [58], and one of the reasons for this could be linked to the high molecular weight employed in these studies. As the main aim of adding chitosan to fluoride salts containing formulas is that of increasing the overall viscosity, high molecular weight chitosan are favored; these, however, have a lower solubility, require an acidic solvent, and have a lower antimicrobial activity [58,59]. Based on these previous findings, we prepared low molecular weight chitosan derivatives and formed fluoride salts using the protonated chitosan as the counterion. The fluoride content in commercial oral health preparations ranges from 200–250 ppm in mouthwashes to 1450 ppm in consumer toothpastes, and up to 5000 ppm in prescription oral products. We successfully loaded fluoride ions on both chitosan and *N*-(2(2,6-diaminohexanamide)-chitosan with CS3H Lys F, able to provide 1450 ppm of F^-^ if dissolved at concentrations lower than 0.1 mg per 100 mL. The fluoride salts of chitosan also showed a higher solubility at neutral pH, allowing for the preparation of solutions at higher concentrations than the parent polymer. All of the polymers tested had a protective effect against acid demineralization, similar to that shown by NaF. Our results confirm the barrier action chitosan can have by forming a protective layer on the tooth surface [57], and also the ability to enhance the effect of fluoride ions by prolonging their contact with the tooth surface. In fact, CS3H Lys F was more effective than the highest dose of NaF, even if it contained a concentration of fluoride ions that was more than 3000 times lower. This is supported by previous findings that indicate that efficient delivery of fluoride has a higher impact on the overall effectiveness of the treatment compared to the dose of fluoride used [60]. This means that the application of fluoridated chitosan has the potential to reduce the therapeutic dose of fluoride required. Previous studies found chitosan of a high and low molecular weight to have a MIC against *S. mutants* in the range of 3–5 mg mL^−1^ [58], while our derivatives have MIC values that are 3- to 5-fold lower. Furthermore, at concentrations as low as 200 µg mL^−1^, our derivatives were significantly more active compared to the starting chitosan, and at the same concertation, these chitosan derivatives were significantly less toxic than CHX (cell viability 8-fold higher). These results suggest that if we load less fluoride ions on the CS3H Lys, we can have satisfactory demineralization protection with negligible cell toxicity and higher antibacterial activity. We further developed mouthwash formulations containing CS3H Lys and CS3H Lys F, and the stability study showed that storage did not have a significant effect on color, odor, solubility of components, and pH of all mouthwash formulations for up to 6 months. Mouthwashes formulated with modified chitosan (CS3H Lys and CS3H Lys F) were more effective at reducing the viability of *S. mutans,* the main cariogenic bacteria, compared to the original chitosan mouthwash. This activity did not change significantly during storage, even at higher temperatures.

## 5. Conclusions

We successfully prepared, for the first time, fluoride salts of chitosan and *N*-(2(2,6-diaminohexanamide)-chitosan. CS3H Lys F showed a higher ability to protect teeth from acid demineralization compared to NaF. The polymer also presented antimicrobial properties and cytocompatibility. The polymer can be further improved by reducing the quantity of loaded fluoride, which would allow for maintaining the protective action and enhance the antibacterial properties. This novel polymer can be formulated in stable mouthwash formulations, and future work could look at its inclusions in toothpastes that are more widely used.

## Figures and Tables

**Figure 1 pharmaceutics-14-00488-f001:**
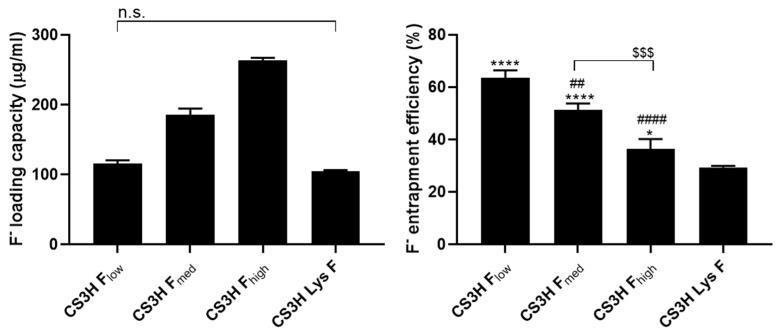
Fluoride ions loading capacity and entrapment efficiency by chitosan polymers. Data are reported as mean ± SD, n = 3. One-way ANOVA returned *p* < 0.0001 for both sets of data, the results of the Tukey’s multi-comparison test are shown in the graphs. For loading capacity, all comparisons returned a *p* < 0.0001 unless otherwise indicated. For entrapment efficiency, * *p* < 0.05, **** *p* < 0.0001 compared to CS3H Lys F, ^##^
*p* < 0.01, ^####^ *p* < 0.0001 compared to CS3H F_low_ and ^$$$^
*p* < 0.001 comparison as specified on graph.

**Figure 2 pharmaceutics-14-00488-f002:**
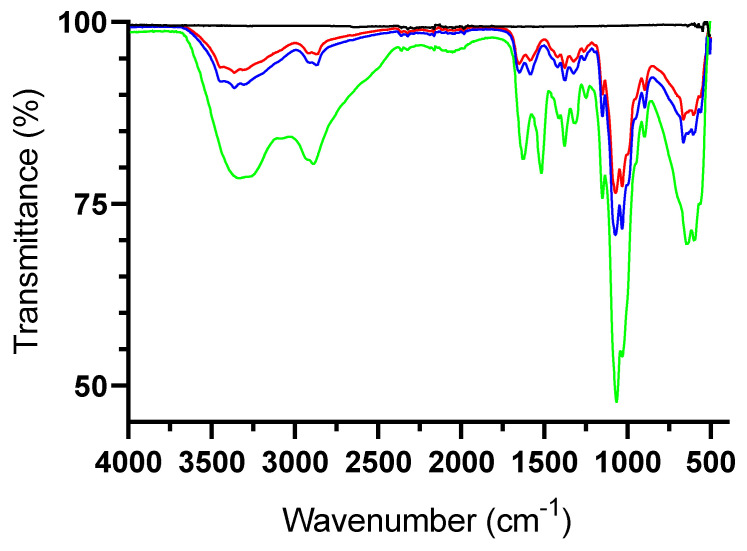
FTIR spectra of NaF (black), CS3H (green), CS3H Lys (blue), and CS3H Lys F (red).

**Figure 3 pharmaceutics-14-00488-f003:**
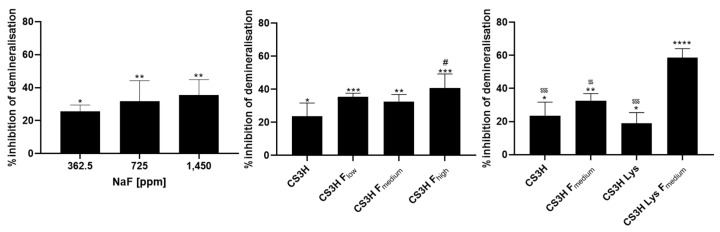
Inhibition of phosphate release by NaF, chitosan, and its derivatives. Data are reported as mean ± SD, n = 3. Data were analyzed by one-way ANOVA followed by Tukey’s multiple comparisons (* *p* < 0.05, ** *p* < 0.01, *** *p* < 0.001 and **** *p* < 0.0001 compared to deionized water; ^#^
*p* < 0.05, compared to CS3H; ^$$^ *p* < 0.01, ^$$$^ *p* < 0.001 compared to CSH3 Lys F_medium_).

**Figure 4 pharmaceutics-14-00488-f004:**
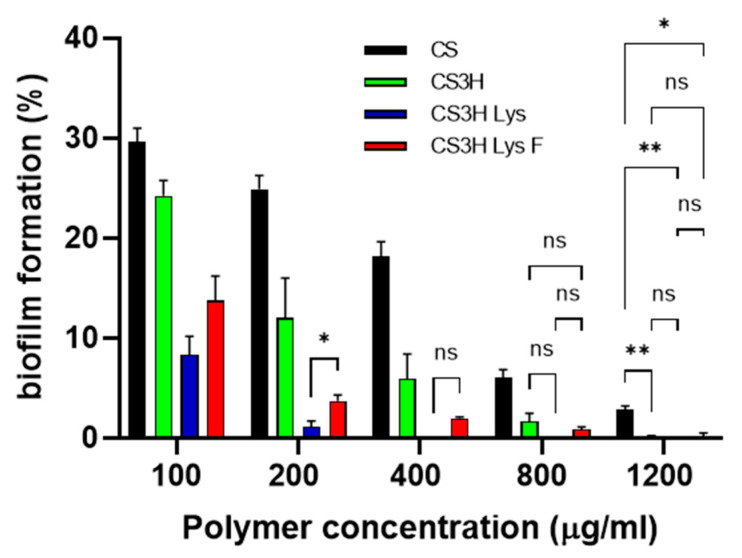
Effect of chitosan polymers on the growth of *S. mutans* biofilms. Data are reported as mean ± SD, n = 6. Data were analyzed by two-way ANOVA (both polymer and concentration effect returned *p* < 0.0001) followed by Tukey’s multiple comparisons; all comparisons between polymers at different concentrations returned *p* < 0.001 unless otherwise indicated in the graph; * *p* < 0.05; ** *p* < 0.001; ns = non-significant.

**Figure 5 pharmaceutics-14-00488-f005:**
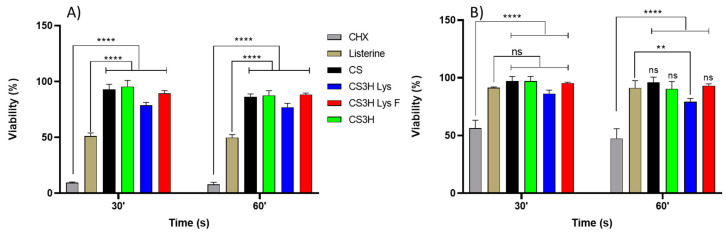
Cytocompatibility test on HGF cells of undiluted mouthwashes (**A**) and mouthwashes diluted 1:1 *v*/*v* with artificial saliva (**B**). Data are given as mean ± SD, n = 4). Data were analysed by two-way ANOVA followed by Dunnett’s multiple comparisons (** *p* < 0.01; **** *p* < 0.0001); ns = non-significant.

**Figure 6 pharmaceutics-14-00488-f006:**
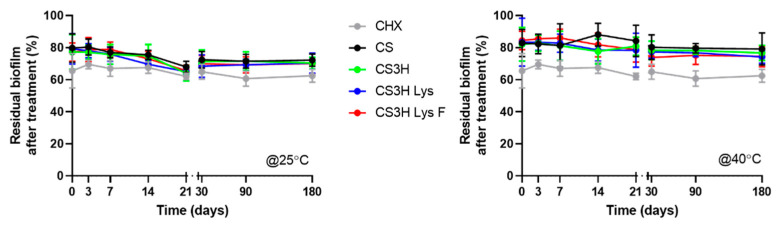
Effect of mouthwashes stored at 25 and 40 °C for up to 6 months on *S. mutans* biofilm formation. Data are reported as mean ± SD, n = 9. Two-way ANOVA: polymer effect *p* < 0.0001, time effect *p* < 0.0001, polymer and time *p* > 0.05.

**Table 1 pharmaceutics-14-00488-t001:** Chitosan-based mouthwash formulation.

Ingredients	Quantity		Function
	Formula I	Formula II	
Glycerin	5 g	5 g	Humectant
Sodium saccharin	450 mg	450 mg	Sweetener
Lutrol 4% *w*/*v*	50 mL	50 mL	Surfactant
Polymer	0.2 mg	0.2 mg	Anti-biofilm
Acetic acid (0.5 M)	1 mL	1 mL	Acidity modifier/co-solvent
Food blue	-	0.2 mL	Coloring agent
Peppermint oil	-	0.25 mL	Flavoring agent
Ethanol	-	20 mL	Co-solvent
Water	to 100 mL	to 100 mL	Vehicle

**Table 2 pharmaceutics-14-00488-t002:** pH dependence of aqueous solubility of fluoridated polymers. Data are reported as mean ± SD, n = 3. One-way ANOVA followed by Tukey’s multi-comparison test was performed to assess the effect of fluoridation with different amounts of fluoride ions on the solubility of CS3H (** *p* < 0.01; *** *p* < 0.001, and **** *p* < 0.0001 compared to CS3H; ^a^ *p* < 0.05 and ^b^ *p* < 0.01 comparing polymers annotated with the same letter). Unpaired, two-tailed *t*-test was performed to assess the effect of fluoridation on the solubility of CS3H Lys (^#^
*p* < 0.05; ^##^
*p* < 0.01; ^###^
*p* < 0.001).

Polymers	Transmittance (%)			
	pH 6.00	pH 6.50	pH 7.00	pH 7.25
CS3H	98.04 ± 0.44	97.72 ± 0.13	69.15 ± 3.5	29.06 ± 3.20
CS3H F_low_	99.34 ± 0.13 **	99.51 ± 0.04 ***	88.32 ± 4.81 ***^a^	72.47 ± 5.64 ****
CS3H F_medium_	99.80 ± 0.20 ***	99.34 ± 0.19 ***	97.30 ± 1.46 ****^ab^	80.21 ± 3.46 ****
CS3H F_high_	99.52 ± 0.25 ***	99.46 ± 0.34 ***	85.80 ± 0.61 ***^b^	72.78 ± 3.91 ****
CS3H Lys	92.88 ± 1.07	92.71 ± 0.26	59.19 ± 7.44	25.22 ± 3.34
CS3H Lys F	88.45 ± 1.89 ^#^	88.67 ± 1.43	86.04 ± 1.16 ^##^	48.5 ± 2.42 ^###^

**Table 3 pharmaceutics-14-00488-t003:** Values of MIC and MBC for the chitosan and modified chitosan against *S. aureus* and *S. mutans*. Data are reported as mean ± SD (n = 4). Data were analyzed by two-way ANOVA followed by Dunnett’s multiple comparisons. No significant difference was observed.

Polymer	*Staphylococcus aureus*		*Streptococcus mutans*	
	MIC (mg/mL)	MBC (mg/mL)	MIC (mg/mL)	MBC (mg/mL)
CS	1.60 ± 0.33	≥3.0	1.50 ± 0.20	≥3.0
CS3H	1.10 ± 0.38	≥3.0	1.30 ± 0.20	≥3.0
CS3H Lys	1.10 ± 0.60	≥3.0	1.40 ± 0.23	≥3.0
CS3H Lys F	1.40 ± 0.23	≥3.0	1.40 ± 0.23	≥3.0

**Table 4 pharmaceutics-14-00488-t004:** Results of the statistical evaluation of absorbance values at 639 nm of different mouthwash formulations over time and at different temperatures. Two-way ANOVA was performed, followed by Sidak’s multiple comparisons test to see the effect of the temperature on the absorbance of the dye in the mouthwash formulation. The data were rerun with Dunnett’s multiple comparisons test to see the effect of the time (ns = not significant or *p* > 0.05; * *p* ≤ 0.05; ** *p* ≤ 0.01).

Polymer	25 °C	40 °C	25 °C vs. 40 °C
CS	T_0_ vs. T_90_ (**) T_0_ vs. T_180_ (*)	T_0_ vs. T_3_ (*)	ns
CS3H	ns	ns	ns
CS3H Lys	ns	ns	ns
CS3H Lys F	T_0_ vs. T_30_ (*)	ns	ns

**Table 5 pharmaceutics-14-00488-t005:** The effect of temperature on the pH of chitosan mouthwash formulation. Data are expressed as mean ± SD, n = 3. Two-way ANOVA followed by Sidak’s test was carried out (* *p* < 0.05 compared to T_0_).

Polymer	Temp	T_0_	T_30_	T_90_	T_180_
CS	25 °C	5.54 ± 0.02	5.56 ± 0.02	5.54 ± 0.02	5.53 ± 0.02
	40 °C		5.53 ± 0.02	5.52 ± 0.01	5.52 ± 0.03
CS3H	25 °C	5.52 ± 0.03	5.53 ± 0.01	5.52 ± 0.01	5.52 ± 0.01
	40 °C		5.50 ± 0.01	5.52 ± 0.01	5.50 ± 0.01
CS3H Lys	25 °C	5.52 ± 0.01	5.52 ± 0.02	5.52 ± 0.02	5.52 ± 0.02
	40 °C		5.48 ± 0.01 *	5.48 ± 0.03	5.48 ± 0.02
CS3H Lys F	25 °C	5.52 ± 0.02	5.54 ± 0.06	5.54 ± 0.04	5.53 ± 0.05
	40 °C		5.50 ± 0.04	5.52 ± 0.06	5.51 ± 0.03

## Data Availability

All data relative to this study are included in the manuscript and Appendix A.

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
