# Peer review of "Harnessing the Antibacterial Properties of Fluoridated Chitosan Polymers against Oral Biofilms"

_pharmaceutics, 2022, doi:10.3390/pharmaceutics14030488_

Round 1

Reviewer 1 Report

Harnessing the antibacterial properties of fluoridated chitosan 2 polymers against oral biofilms is an interesting study,

  • Surface morphology should be presented before and after loading fluoride.
  • FTIR spectra also presented with blank fluoride and chitosan.
  • FTIR model should be mentioned in the methods section.
  • Spectrophotometer model should be mentioned.

Author Response

Reviewer #1

Harnessing the antibacterial properties of fluoridated chitosan 2 polymers against oral biofilms is an interesting study,

The authors would like to thank the reviewer for taking the time to assess our work and for suggesting ways to improve it.

  • Surface morphology should be presented before and after loading fluoride.

Since we are working with a soluble polymer no significant change in surface morphology is expected, SEM coupled with EDX has been carried out to address this point and images are reported in the supplementary information. We found that the morphology was not changed however in addition to an homogeneous presence of F in the bulk of the polymer, some deposits of NaF were also present on the surface. 

  • FTIR spectra also presented with blank fluoride and chitosan.

Figure 2 has been modified to include the spectra of NaF and CS3H

  • FTIR model should be mentioned in the methods section.

This is stated in section 2.4 as “Varian FTIR spectrophotometer (Agilent Technologies, Stockport, UK)”

  • Spectrophotometer model should be mentioned.

Two different spectrometers have been used whether a plate or a cuvette was required, these are reported in section 2.5 as “Nicolet e-100 spectrophotometer (Thermo Fisher Nicolet Corp., UK) using a quartz cell” and in section 2.6 as “SpectraMax i3x, Molecular Devices, Berkshire, UK”. 

Reviewer 2 Report

The authors investigate a new formulation against oral biofilms related to dental caries. Extending their previous work, fluoridated chitosan polymers are used here, the new formulation is thoroughly characterized, e.g., by FTIR spectroscopy and solubility data, and it is shown to have a lower minimum inhibitory concentration and negligible cell toxicity. These findings are important, timely and of interest to the readership of Pharmaceutics. Moreover, the paper is well written, the results seem to be robust and I could not find any flaw. Therefore, I recommend to accept the paper.

Author Response

The authors would like to thank the reviewer for the kind words about our work.

Reviewer 3 Report

The manuscript dealing with the antibacterial properties of fluoridated chitosan polymers to be used for mouthwashes is well written. However, the novelty of the manuscript is limited. There are other studies proving that chitosan and fluoridated chitosan solutions improved the dentin resistance to erosion and provide teeth surface protection from erosion.

The authors should include the ionic exchange capacity of the polymers and based on it to add the  amount of fluoride ions to the polymers. Hence, the efficiency of fluoride ion entrapment is not relevant and can be deleted (Fig. 1). Of course, a high concentration of fluoride, well above the ion exchange capacity of polymer leads to a low efficiency.

Also, the FTIR spectra (Fig. 2) should be moved in Supplementary Materials because is obvious that an electrostatic interaction cannot lead to observable modification in the FTIR spectrum.

A description of N-(2(2,6-diaminohexanamide)-chitosan (CS3H Lys) and CS3H obtaining in brief, especially that for the first the reference given is only a submitted manuscript, should be provided.

The authors need to better point out the novelty of the research, for instance the use of fluoride of CS3H Lys.

Author Response

Reviewer #3

The manuscript dealing with the antibacterial properties of fluoridated chitosan polymers to be used for mouthwashes is well written. However, the novelty of the manuscript is limited. There are other studies proving that chitosan and fluoridated chitosan solutions improved the dentin resistance to erosion and provide teeth surface protection from erosion.

To the best of our knowledge previous work has only looked at supplementing chitosan solution with fluoride salts, no ion loading directly into the polymer has been reported. This has the potential to make fluoride ions available on site but not available for systemic absorption, however we have not added this claim as this hypothesis has not been tested. 

The authors should include the ionic exchange capacity of the polymers and based on it to add the  amount of fluoride ions to the polymers. Hence, the efficiency of fluoride ion entrapment is not relevant and can be deleted (Fig. 1). Of course, a high concentration of fluoride, well above the ion exchange capacity of polymer leads to a low efficiency.

Ion exchange capacity determination is generally carried out on solid membranes, while here we studied soluble chitosan derivatives. Furthermore the use of a strong acid such as sulphuric acid, as required in the protocol for the determination of the ion exchange capacity of membranes, would lead to the degradation of our polymers.

Also, the FTIR spectra (Fig. 2) should be moved in Supplementary Materials because is obvious that an electrostatic interaction cannot lead to observable modification in the FTIR spectrum.

As another reviewer has asked for the addition of NaF and chitosan FTIR spectra the figure has been kept in the manuscript.

A description of N-(2(2,6-diaminohexanamide)-chitosan (CS3H Lys) and CS3H obtaining in brief, especially that for the first the reference given is only a submitted manuscript, should be provided.

The description of the synthesis of CS3H Lys has been added to the supplementary information.

The authors need to better point out the novelty of the research, for instance the use of fluoride of CS3H Lys.

We have highlighted in the introduction that the  preparation of CS3H Lys F has here been done for the first time.